# Machine Learning Characterization of Ictal and Interictal States in EEG Aimed at Automated Seizure Detection

**DOI:** 10.3390/biomedicines10071491

**Published:** 2022-06-23

**Authors:** Gaetano Zazzaro, Luigi Pavone

**Affiliations:** 1C.I.R.A.—Italian Aerospace Research Centre, Via Maiorise s.n.c., 81043 Capua, Italy; g.zazzaro@cira.it; 2I.R.C.C.S. Neuromed, Via Atinense, 18, 86077 Pozzilli, Italy

**Keywords:** data mining, electroencephalogram, epilepsy, false-alarm rate, intracranial EEG, *k*-nearest neighbor, machine learning, signal processing, seizure detection

## Abstract

Background: The development of automated seizure detection methods using EEG signals could be of great importance for the diagnosis and the monitoring of patients with epilepsy. These methods are often patient-specific and require high accuracy in detecting seizures but also very low false-positive rates. The aim of this study is to evaluate the performance of a seizure detection method using EEG signals by investigating its performance in correctly identifying seizures and in minimizing false alarms and to determine if it is generalizable to different patients. Methods: We tested the method on about two hours of preictal/ictal and about ten hours of interictal EEG recordings of one patient from the Freiburg Seizure Prediction EEG database using machine learning techniques for data mining. Then, we tested the obtained model on six other patients of the same database. Results: The method achieved very high performance in detecting seizures (close to 100% of correctly classified positive elements) with a very low false-positive rate when tested on one patient. Furthermore, the model portability or transfer analysis revealed that the method achieved good performance in one out of six patients from the same dataset. Conclusions: This result suggests a strategy to discover clusters of similar patients, for which it would be possible to train a general-purpose model for seizure detection.

## 1. Introduction

### 1.1. Background

Epilepsy is a chronic neurological disease that affects around 50 million people worldwide and is characterized by recurrent seizures, which are unpredictable and severely affect the quality of life of patients [1]. Epilepsy monitoring and diagnosis is generally performed by recording electroencephalogram (EEG) signals, either from the scalp or intracranially (iEEG). In most cases, EEG monitoring is conducted continuously for a few days in order to confirm the diagnosis of epilepsy and to anatomically localize the epileptic focus, especially in patients with drug-resistant epilepsy or in those with unclear diagnosis [2]. EEG signals acquired during long-term monitoring are visually reviewed and analyzed by epileptologists, but this work is extremely time consuming and labor intensive because of the large amount of EEG recordings available for each patient [3,4,5,6]. Furthermore, the diagnosis of epilepsy is a complex task that requires accurate documentation by the patient himself or by his relatives, but this information is often incomplete and not accurate [7]. Therefore, the development of computerized methods able to automatically detect seizures in the EEG recordings could help clinicians during diagnosis, speeding up and even improving the process of seizure identification. Furthermore, such methods could be included into closed-loop intervention systems for epilepsy treatment, especially in those patients who do not respond to antiepileptic drugs (about 30% of people with epilepsy). In this case, patients could be alerted of an incoming seizure or few seconds after its arising, and this could potentially improve the quality of their life and the associated risk when they experience a seizure. For example, in closed-loop stimulation systems, such algorithms, when a seizure is detected, can trigger electrical stimulation in the area where the seizure is detected and abort the seizure. This procedure is often the only option for many patients, but it is invasive for the patient. Besides a very high sensitivity (true-positive rate) necessary to detect all the seizures, a very low false-positive detection rate is a key achievement that such systems should ensure in order to minimize the invasiveness for the patient, because the higher the number of false alarms is, the higher the number of electrical stimulations delivered is, and the more numerous the side effects for the patient are. Currently available EEG-based online seizure detectors used in video-EEG monitoring units are associated with an unacceptable rate of false alarms for ambulatory patients, varying between 0.1 and 5 per hour [8]. 

A typical seizure detection system consists of three main steps, including the pre-processing of brain signals (band-pass filtering, denoising, artifact removal), feature extraction from the preprocessed input signal, feature selection, and classification using machine learning or deep learning models to detect seizures. Although the performance of these systems is influenced by the preprocessing step and by the extracted features, the improvement of the performance of the classification models is the key challenge to develop very accurate and efficient seizure detection methods. In recent years, intense research has been conducted by various research groups, also thanks to the increasing interest in machine learning techniques. 

### 1.2. Related Works

Various machine learning approaches have been proposed in order to develop efficient and reliable automated seizure detection methods, such as extreme machine learning [9,10], recurrent Elman network [11], support vector machine [12,13], random forest [14,15], artificial neural networks [16], linear discriminant analysis [17], and k-nearest neighbors [18,19]. 

In [9,10], an optimized sample entropy algorithm combined with extreme machine learning was used to identify the EEG signals containing seizures. The authors of [11] used three types of entropy measures and two different neural network models, recurrent Elman network and radial basis network, to classify normal and epileptic EEGs. In [12], fuzzy entropy and sample entropy, together with a grid optimization method, were used to train support vector machines (SVMs). SVMs were also used in [13] in combination with log energy entropy on band-passed EEGs. A random forest classifier was used in [14] in combination with various features both in the frequency and time domains extracted from previously selected EEG channels, while in [15], the same kind of classifier was used in combination with various features from the time domain, the frequency domain, and entropy-based features. The authors of [16] classified seizure and non-seizure signals using extracted frequency and amplitude information passed to artificial neural nets. Furthermore, a linear discriminant analysis classifier was used in [17] to classify seizure and non-seizure EEGs using features extracted with a wavelet-based technique. Finally, in [18,19] *k*-nearest neighbor (k*-*NN) algorithm was used in combination with various features extracted from the statistical domain or from the frequency, entropy or mathematical domain, respectively. None of these studies provided a comprehensive analysis of the false alarms, and only few studies also analyzed the EEG signals acquired in the interictal phase, but only the preictal and ictal phases were considered for the creation of the seizure detection model.

### 1.3. Hypothesis and Specific Aims

Since the main limitations of existing seizure detection methods are the unacceptable numbers of false alarms [8] and the high level of inter-patient heterogeneity, in this study, we propose a seizure detection methodology based on data mining and a k*-*NN algorithm with which we deeply investigated the performance of the method in terms of the quantification of the number of false alarms. Furthermore, we conducted a portability analysis of the method on different patients of the same dataset in order to investigate if the method is generalizable to different patients with different clinical characteristics. Thus, the specific aims of this study were:
To develop an automated seizure detection method that is able to detect seizures with high accuracy and very low false-positive rates;To evaluate, qualitatively and quantitatively, the false-alarm rate on interictal data;To test the portability of the method by testing the model on different patients.


## 2. Materials and Methods

In this section, we describe all the steps and the procedures carried out in order to achieve the research objectives, including the raw data description and the analysis methodology design, with the details of the selected machine learning algorithm and of the software tool used.

### 2.1. Methodology

The proposed approach was carried out in five main steps. First, the raw EEG data from one patient were preprocessed by applying a 50 Hz notch filter to remove line noise and by applying a band-pass filter to the noise-free EEG data. Then, feature extraction from band-passed EEGs was performed, and 1080 features were extracted from each band-passed signal. These two steps were performed using a custom-made software conceptualized, designed, and developed by us, Training Builder (TrB) software [20], a multipurpose and extendable software tool that allows one to analyze EEG signals using the sliding window paradigm [21]. Afterwards, feature selection was performed using the Information Gain and Pearson Correlation filters. Then, a machine learning approach using a k-nearest neighbor (k-NN) algorithm was applied on preictal, ictal, and interictal EEG recordings. Finally, the selected k-NNs were tested on the EEG recordings of five other randomly selected patients from the same dataset to test the portability of the model to other patients. The complete analysis pipeline is shown in Figure 1.

### 2.2. Dataset

The EEG data of one patient (number 17) from the Freiburg Seizure Prediction EEG database (FSPEEG) [22], which contains the EEG recordings of 21 patients with epilepsy acquired at a 256 Hz sampling rate, were used. For each patient, the EEG recordings were acquired in three different time periods: during the seizure (ictal phase), just before the ictal phase (preictal phase), and between two subsequent seizures (interictal recordings). These EEG recordings were acquired from six EEG channels, with three being located near the epileptic focus (“InFokus” channels) and the remaining three being located outside the seizure focus (“OutFokus” channels). In the present study, we selected ictal and interictal EEG data from patient number 17, because five long-lasting epileptic seizures (average seizure duration of 86.16 s) were available. 

### 2.3. EEG Preprocessing

Each EEG recording was first preprocessed; then, a list of features was extracted from the preprocessed EEGs using TrB software. First, a 50 Hz notch filter was applied to EEG data in order to remove line noise due to signal acquisition and digital conversion. Afterwards, a band-pass filter was applied on the noise-free EEG data in order to extract the signal content in six different frequency bands corresponding to well-known oscillations of brain activity, namely, (8–13) Hz (α), (13–21) Hz (β1), (21–30) Hz (β2), (30–40) Hz (low γ), (40–70) Hz (medium γ), and (70–120) Hz (high γ). Finally, a sliding window paradigm was applied to the noise-free, band-passed EEG data by setting two temporal parameters: L, which represents the time length of the analysis window, and S, which is the time shift of the window which slides on the signal. Therefore, every S seconds, the previous L seconds of the signal were analyzed (L×256 points, where 256 is the sampling frequency); considering the 6 EEG channels and 6 frequency bands, L×256×6×6 points were processed every S seconds. Finally, numerous features were extracted from these EEG segments for all the six EEG channels, the InFokus (electrode numbers 1, 2, 3) and the OutFokus (electrode numbers 4, 5, 6). These features were both univariate, i.e., computed with respect to the “actual” signal, and bivariate, i.e., computed with respect to two signals, the “actual” one and another one (alternatively, the “previous” signal or the origin signal).

### 2.4. Feature Extraction

In the feature-extraction step, the pre-processed time-series data were analyzed by applying the sliding window paradigm available in the TrB. The aim of this step was to extract a set of variables (features, attributes, or characteristics) as the electrodes and frequency bands varied in each window, that is, in each sub-series of length L seconds that slid every S seconds. The features, shown in Table 1, were computed with the TrB tool using a set of algorithms whose description can be found in [20]. In particular, seismic evaluators (IDs 6–11 in Table 1) were calculated because of the analogy between earthquakes and epileptic seizures [23,24].

The temporal parameters of the sliding window were fixed following the studies conducted in [25], namely, L=2 s and S=1 s. This means that, every second (S=1), two seconds (L=2) of the previous signal were analyzed, and from these records, the features were calculated.

From the analysis of approximately 40 h of recordings of patient No.17 using the TrB tool, we obtained a dataset (called D) with 136,341 instances described by 1081 calculated features. In particular, 135,909 instances were tagged with “N” (no seizure), because they were extracted from signals without seizures (preictal and interictal), and 432 were tagged with “Y” (seizure), because they were extracted from signals with seizures (ictal). The extracted features were univariate or bivariate [26]. The bivariate ones were calculated with respect to two different methods: method A (MA), which considered a signal with the same length but translated backwards by S seconds (previous signal) as the second signal; method B (MB), which considered the origin (null vector) as the second signal.

Table 1 also shows the count of each feature in the dataset. Since there were 6 electrodes (3 InFokus and 3 OutFokus) whose signals were filtered into 6 different frequency bands, each univariate characteristic was calculated 36 times, whereas each bivariate feature was calculated 72 times (36 using method A and 36 using method B). This resulted in 1080 features, to which the target class, “Actual *YN*”, was added.

### 2.5. The k-NN Algorithm

The learning algorithm chosen for the data-modeling phase with the aim of detecting seizures was k-nearest neighbor (k-NN) [27]. k-NN is one of the most used machine learning techniques for data mining, showing very high performance in many applications [28], such as in satellite scene classification, handwritten digit identification, fraud detection, ECG-pattern discovery, and in the detection of new COVID-19 [29] from human genome sequences. k-NN is often successful where the classes are not linearly separable because the decision boundary is very irregular [30].

The k-NN algorithm is very simple, and it is based on learning by analogy, whereby a new point is classified considering its closest (k) neighboring points. It is founded on the hypothesis that similar inputs are usually related to similar outputs. In the simplest case where k=1, the class of the instance that is most similar (or close) to the new point is assigned as the output class. If k>1, then the output class is assigned to the new vector computing the majority of the classes of the k-nearest points (with k odd). The k number of the nearest neighbors can be obtained following, for example, the study [31].

The algorithm computes the distances between each point in the test set and all the points of the training set in order to obtain the nearest neighbor (NN) list. “Analogy” (or “closeness” or “nearness”) is usually defined in terms of Euclidean distance, but other choices are possible [32]. The general scheme of the algorithm is shown below [33] (Algorithm 1).


**Algorithm 1**

Let *k* be the number of the nearest neighbors. Let *A* be the training set and *B* be the test set.for each $b=\left(b,y^{\prime }\right)\in B$ doCompute d(b,a), the distance between *b* and every point (a,y)∈ASelect $\;A_{b}\subseteq A$, the set of *k* closest training points to *b*

$y^{\prime }=arg\underset{t}{\max}\underset{\left(b_{i},y_{i}\right)\in A_{b}}{\sum }I(c=y_{i})$

end for


Once the NN list is found, test point b is labeled based on the majority class of its nearest neighbors (majority voting), where c in $y^{\prime }$ is a class label; $y_{i}$ is the class label for one of the nearest neighbors; and I(⋅) is a characteristic function that is equal to 1 if its argument is true and 0 if its argument is false.

A very common problem of the k-NN algorithm is the so-called curse of dimensionality [34]. The curse of dimensionality means that Euclidean distance is unhelpful in very high dimensions because all points in the training set are almost equidistant to the search point. Hence, it is advisable to preliminarily reduce the number of variables by applying, for example, a technique for feature selection or feature extraction [35].

Finally, it is useful to underline that no classification model is trained using the k-NN algorithm; to classify a new instance, all its distances are calculated with all the elements of the training set. For simplicity, the algorithms are still called classifiers or classification models.

### 2.6. Performance Measures

All the performance measures of a model, such as a classifier, are based on the confusion matrix (CM) [33]. A very common form of CM, which we used in our work, is shown in Figure 2. The elements tagged with “N” are the negative elements of the target class in the dataset, and they represent, in our case study, the majority class. The elements tagged with “Y” are the positive elements of the target class in the dataset, and they form the minority class.

“TN” elements are the true-negative elements of the CM, that is, the negative elements of the dataset correctly classified as negative by the model. “TP” elements are the true-positive elements of the CM, that is, the positive elements of the dataset correctly classified as positive by the model. “FN” (false negative) elements are the positive elements of the CM that are misclassified as negative by the model. Finally, “FP” (false positive) elements are the negative elements of the CM that are misclassified as positive by the model. Sum S of all the elements of the CM is the size (cardinality) of the dataset, that is, the number of its elements. The classifier with the best performances has $\frac{TP}{P}=1$ and $\frac{TN}{N}=1$, because all the positive elements and all the negative elements are correctly classified.

A very common performance metric is the classification accuracy (CA), which measures how good a model is at correctly predicting both positive and negative cases:(1)$$Classification\;Accuracy=\left(CA\right)=\frac{TP+TN}{TP+TN+FP+FN}=\frac{TP+TN}{P+N}=\frac{TP+TN}{S},$$

The other performance metrics [33] used in this work are reported in Table 2.

### 2.7. Hold-Out Method and k-Fold Cross-Validation

The hold-out method [33] is a very common strategy in machine learning mainly aimed at providing a useful framework for dataset splitting and designing in order to train a model and evaluate its performance and to avoid the overfitting problem [36]. According to this strategy, the full dataset of selected features, labeled with “Y” or “N”, was partitioned into two random disjointed sets; the first one, called training set, was used for inducing the model by applying a learning algorithm, while the second one, called test set, was used for testing the trained model by using performance metrics (Table 2). In most cases, the subdivision percentages can vary.

The hold-out method is the simplest cross-validation method. k-fold cross-validation is a resampling method and uses k different portions of the data to train and test a model on k different iterations. According to this statistical method, the original sample was randomly partitioned into k equally sized subsamples. Of the k subsamples, k−1 subsets were used as training data, and the remaining single subset was used as validation data for testing the model. This process was then repeated k times, with each of the k subsets used exactly once as the validation data. Moreover, in our case study, each subdivision of the dataset was stratified, which meant that it retained the distribution of the target class. The k results, that is, the k sets of performance obtained, could then be averaged to obtain a single estimation that generalized the behavior of the algorithmic approach used to train the model. In this work, k was fixed to 10.

### 2.8. Class Imbalance Problem

In k-NN, as well as in most machine learning algorithms, the classification performance is significantly impacted by the unbalanced class. Data suffer the class imbalance problem when the target class distributions are highly imbalanced. Although some authors have pointed out that a strong class imbalance is not always a problem in learning from data [37], the classifier-training phase is often profoundly conditioned by the majority class (negative class); even if the models can have high accuracy (total amount of correct classifications), when viewed more closely, they show a low TPR and low Precision for the classification of positive elements (rare events).

Imbalance can be characterized by the imbalance ratio (IR), here calculated as the ratio between the number of negative elements (N; majority class) and the number of positive elements (Y; minority class) in the database. In particular, after the feature-extraction phase (Section 2.4), full dataset D, obtained using the TrB tool, had, as said, 135,909 instances tagged with N because they were extracted from signals without seizures and 432 instances tagged with Y because they were extracted from signals with seizures. D had IR=#N#Y=135,909432=314.6. For this reason, D suffered the class imbalance problem.

## 3. Data Preparation

The data preparation step included the application of feature-selection algorithms for dataset dimensionality reduction, the hold-out method for dataset splitting, and the application of our technique for overcoming the class imbalance problem. Therefore, this section describes all the steps to obtain the final datasets for the forthcoming modeling phase with the k-NN algorithm for classification.

### 3.1. Feature Selection

The feature-selection phase was aimed at reducing the number of variables (Table 1) that were calculated by the TrB tool by setting the temporal parameters of the sliding window. In this step, we chose to evaluate the worth of an attribute by measuring its Information Gain with respect to the class. In this way, we wanted to favor those variables having a greater expected amount of information (reduction in entropy). 

The Information Gain (IG) [33] of an attribute X was calculated using Formula (2):(2)$$\begin{array}{c} IG\left(Class,X\right)=H\left(Class\right)-H\left(Class|X\right),\\ H\left(X\right)=-p\left(x\right)\sum p\left(x\right)\ln p\left(x\right), \end{array}$$
where H(X) is the Shannon Entropy [38] of X.

We assigned a score, or rank, to each attribute using the IG; these scores were then sorted in descending order, and the highest variables on the list were selected. The threshold for IG selection was set to 0.85, so all features having an IG<0.85 were rejected. In this way, 41 features survived. Note that the data were normalized before ranking. After this first selection, a second filter was applied, which evaluated the worth of an attribute by measuring the Pearson Correlation Coefficient between it and the target class. The Pearson Correlation Coefficient (PCC), whose values always range between −1 and 1, is the covariance of the two variables divided by the product of their standard deviations. All the features having −0.75<PCC<0.75 were deleted. The applications of these two filters reduced the number of the features from 1080 (Table 1), as calculated by the TrB tool, to 22. These were divided into 7 categories (CE-MA, CE-MB, JE-MB, KC, LD-MA, MI-MA, and SP). 

The distributions and the frequencies (in brackets) of the 22 survived features, together with the electrode (E) number and the frequency band (B), are reported in Table 3. The sum of the numbers in the brackets for each row of Table 3 is 22. 

The two applied filters excluded the features calculated for the OutFokus electrodes and for frequency bands lower than 40 Hz. Moreover, most of the features selected concerned those deriving from the information theory [20,38,39].

### 3.2. Test Plan

In our case study, the partitioning of the hold-out method was carried out by considering the different phases of the EEG signal. Table 4 shows the list of the datasets used in our analysis useful to select the best k-NN. Testing the models on the IKTAL and INTERIKTAL sets allowed us to select the best performing classifiers with respect to the detection of seizures and the reduction in false alarms (false positives).

### 3.3. Optimized Matthews Undersampling Technique

As specified in Section 2.8, the complete dataset D of patient No.17 had $IR=\frac{\#N}{\#Y}=\frac{135,909}{432}=314.6$. If the sets for the tests (IKTAL and INTERIKTAL) were separated from the complete dataset, then remaining set T (T=D−(IKTAL∪INTERIKTAL)) had an $IR=\frac{\#N}{\#Y}=\frac{95,798}{337}=284.3$.

In the literature, several techniques have been proposed to deal with the problem of learning from imbalance datasets; mainly, they are based on training data resampling algorithms [40,41], but also on cost-sensitive approaches or on ensemble methods [42]. In our study, in order to overcome the class imbalance problem, a random undersampling strategy, which we called Optimized Matthews Undersampling (OMU) method, was applied to resample the T set. In particular, this strategy is based on n=10 different random undersamplings $T_{i}$ of T, where each set $T_{i}$ always has the same (all) 337 records as the minority class (tagged with “Y”), whilst the majority class of each $T_{i}$ has 337×IR (IR=1, 2, 5, 10, 20, 25, 50, 100) records (tagged with “N”). Moreover, if $T_{i}$ and $T_{j}$ are two different undersampled sets of T (i,j=1,…,10 and i≠j), $T_{i}$ and $T_{j}$ may also have elements of the majority class in common because of the random extractions. This method is useful for trying to minimize the loss of information of the majority class, which can often be a side effect of the “simple” undersampling method.

In order to choose the final balanced training set, $T_{i}$ (i=1,…,10), 10 different k-NNs with k=1 (each of them obtained for a different $T_{i}$) were 10-fold cross-validated, and the $T_{i}$ for which the 1-NN had the highest Matthews Correlation Coefficient (MCC) (Table 2) was selected. The MCC, that is, the correlation coefficient between the observed and predicted binary classifications, was fixed, because several authors [43,44,45] have pointed out that in the case of unbalanced datasets, the MCC is the best choice for the selection of the classifier. Table 5 reports the values of the MCC of the selected training sets. In particular, for each IR, its maximum values are shown.

In order to recap, OMU was applied to T 8 times (one for each IR) obtaining 8 different training sets. To these 8 sets, set T was added (T preserves the original imbalance ratio, IR=284.3). Table 5 also shows the MCC of the 1-NN trained on the T set (IR=284.3) that had the lowest MCC value.

## 4. Results

The outcomes of data classification depend on the selection of parameters of the k-NN algorithm and, in our case, on the k parameter and on the IR value, which are related to how the training data were prepared. In this section, we use the symbol NN(IR,k) to indicate that the k-NN algorithm was applied starting from the dataset with the IR (imbalance ratio) and considering the k-nearest neighbors (the k parameter varied in {1,3,5,7,9,11,13,15,17,19,27}). The k-NN results are reported considering the k-fold cross-validation (k=10) and the tests on the ictal and interictal phases.

### 4.1. Performance by 10-Fold Cross-Validation

The 10-fold cross-validation guided us in selecting the best models (i.e., the optimal hyperparameter set of the k-NN algorithm), providing a criterion for comparing the calculated performance metrics. 

In the figures in the current section, but also in the figures in Section 4.2, each curve refers to a different training set obtained by setting a different imbalance ratio (IR) in the OMU technique (Section 3.3). Figure 3, Figure 4, and Figure 5 describe, respectively, the trends of Recall on Y, Precision on Y, and Recall on N as the k number of the nearest neighbors varied.

The curves show that IR and k had an impact on the performance of the classifiers, as expected. In particular, the lower the IR was, the more the k-NN was able to correctly classify the examples belonging to the positive class (Figure 3) and thus the more it was able to detect records that contained seizures. Dually, the higher the IR was, the higher the correct classification of negative instances was (Figure 5), i.e., the more the k-NN was able to correctly classify the records tagged with N. Moreover, Recall on Y (TPR) had a very low variability as k varied, except for the curves of IR=Original=284.3 and IR=100, whilst Recall on N (TNR) was almost always constant as k>1 varied, except for the curve of IR=1, which showed a wider variability of the metric for every k.

In order to obtain a compromise among all these situations and to select the best classification models, we fixed some ad hoc thresholds for the analyzed performance metrics. Considering a threshold of 0.92 for Recall on Y (Figure 3), the model that exceeded it was NN(IR,k), with IR≤25 and k≥1. Then, considering a threshold of 0.95 for Precision on Y (Figure 4), the survived models were:
NN(IR,k), with IR≤5, k≥1;NN(IR,k), with 10≤IR≤25, k≥3.


Furthermore, considering a threshold of 0.998 for the TNR (Figure 5), the resulting models were:
NN(IR,k), with IR=100, Original, k≥1;NN(IR,k), with IR=25, 50, k≥3;NN(IR,k), with IR=20, 5≤k≤19.


Moreover, the results shown were consistent with the *MCC* values shown in Table 5 for k=1.

Finally, in order to recap, in Figure 6, we show as green cells all the 10-fold cross-validated models that met all the above-mentioned performance criteria.

### 4.2. Performance on IKTAL and INTERIKTAL Tests

All NN(IR,k)s were compared using the performance measures calculated on the IKTAL and INTERIKTAL sets. In particular, the best models selected by 10-fold cross-validation (Figure 6) were tested on these two independent sets.

The IKTAL set was formed by recording No. 122, containing only the preictal phase, and recording No. 123, containing an ictal phase that lasted about 94 s, after a short phase of continuation of the preictal phase. The INTERIKTAL set was formed by 10 h of registrations whose samples were all tagged with N.

The curves in Figure 7 and Figure 8 show that the IR and k also had an impact on the results of these new tests.

The best models, tested on IKTAL and evaluated with respect to Recall (TPR), were those obtained by considering the sets with the lowest IR; the more balanced the training set was, the higher Recall on IKTAL was. Moreover, the TPR varied very little as k varied, especially for low values of IR. Furthermore, Recall on N (TNR) reached the value of 1 (about 100% of correctly classified negatives) for each k>1 and for each IR>5. From the curves in Figure 7, very high performance was achieved by NN(IR,k), for each k, if IR≤20. These obtained rules further reduced the number of green cells in Figure 6.

The test on the INTERIKTAL set was carried out considering the total number of false-positive (FP) elements (Figure 8). A model with a lower number of FP elements was preferred, in order to avoid false detections as much as possible. From the curves in Figure 8, we deduced that the higher the IR was, the lower the number of negative elements wrongly classified as positive by the models was. This rule confirmed what was achieved in the other tests. However, all k-NNs had a very low number of FP elements when tested on INTERIKTAL. The maximum number of FP elements was 446 and was obtained by NN(1,1). Moreover, by quantifying the obtained rule by which, as the IR increased, the FP number decreased in INTERIKTAL recordings, NN(284.3,19), whose training set preserved the original unbalanced class distribution, only had 10 false positives.

Putting together the rule obtained in the IKTAL set, i.e., that IR≤20, and the one obtained in the INTERIKTAL set, i.e., to have the highest IR, the optimal model could be obtained along the green curve (FP20) in Figure 8, that is, for a k in [5,19] that minimized the number of FP elements. For k=5, 7, 9, NN(20,k) only had 38, 39, and 39 false positives, respectively. Finally, we decided to choose k=7 because NN(20,7) had the highest Recall (TPR) on the IKTAL set.

### 4.3. The Final Model

The performances of all k-NNs were compared as the two parameters IR and k varied. Since the IR varied in 9 ways and k in 11 ways, we had IR∗k=9∗11=99 different k-NNs, each of which was tested three times (by 10-fold cross-validation, and on the IKTAL and INTERIKTAL sets). The 297 (99 k-NNs ∗3 tests) results were compared considering their confusion matrices, Recall values on Y and on N, Precision values, and AUCs, whilst the performances of the k-NNs on INTERIKTAL were compared by only considering the FP elements.

As mentioned in Section 4.2, the final model chosen was model NN(20,7). Figure 9 reports all the performances of the selected model with respect to the 10-fold cross-validation, to the IKTAL set useful for the detection of seizures, and to the INTERIKTAL set useful for an optimal quantification of FP elements. Moreover, the confusion matrices are also reported. Obviously, since the INTERIKTAL test file only had negative elements, the performance metrics could not be calculated on it. Figure 9 also shows the very high performance of the selected model, both for the detection of seizures and for the reduction in FP elements. NN(20,7) was chosen trying to find the best trade-off among the high-lined results of Section 4.1 and Section 4.2. The objective pursued was, therefore, to have the maximum possible number of true positives in the IKTAL phase, trying to limit as much as possible the FP elements in all the analyzed phases, especially in the INTERIKTAL one, in order to avoid the detection of unreal seizures.

Therefore, the best classifier showed a very low FP number and a very high TP number. In particular, in the IKTAL phase, the final model had FP=0 and TPR=93.68% (only six FN elements). Moreover, NN(20,7) only had 39 false-positive elements, which emerged during the test on the INTERIKTAL set.

### 4.4. Error Analysis 

In order to understand the results achieved by the NN(20,7) model in the detection of seizures, an error analysis was also performed. Figure 10 shows the trend over time of the Kolmogorov Complexity (KC; in E1, B40) of the IKTAL file (we chose the KC for graphic reasons only). The preictal phase is shown in green, the ictal phase in red. Every point refers to a second of registration (L=2 and S=1). Points classified by NN(20,7) as negative elements are indicated with a small circle, while those classified as positive are indicated with a larger circle. Moreover, Figure 10 also shows a detail of IKTAL registration in the box, in which six positive points classified as negative (FN elements) by the model can be seen. Four out of these six classification errors were at the beginning of the ictal period, and this can be interpreted as a 4 s delay of the method in detecting the seizure. This mistake could be due to a somewhat fuzzy start of the seizure. However, more likely, it may also be due to the tagging of the incipit of the crisis, which is very subjective and highly depends on the epileptologist that manually marks the seizures in the EEG traces.

The remaining two classification errors, also consecutive, were positioned at seconds No.21 and No.22 of the seizure, when the seizure was already ongoing. Therefore, it can be concluded that the seizure was correctly detected by NN(20,7).

Figure 11 describes the trend over a time of 10 h of interictal recordings, where, obviously, there were no elements tagged with Y. In total, 39 false-positive elements (large circles) were classified by NN(20,7).

In the lateral box of Figure 11, the detail of the agglomeration of 15 of the 39 points erroneously classified as positive is shown, and the list of the time points where they were detected is shown in Figure 12. These 15 points, but also the remaining points in the graph, were not consecutive, and, at most, there were two consecutive points. These errors were almost always located at the spikes of the Kolmogorov Complexity curve.

Finally, we tried to inspect the signal from E1 in frequency band B40 in order to understand where these false positives came from, with a particular focus on the time period in which the model detected a “cluster” of consecutive or semi-consecutive false positives (see Figure 11). We found that the signal in that time period was most likely affected by artifacts, as shown in Figure 13.

### 4.5. Portability Analysis

In order to test if the NN(20,7) chosen for the seizure detection of patient No.17 could be used to detect the seizures of other patients, a portability analysis (intended as a study of the conditions, or limitations, for the transfer, or exportability, of the learning model from one patient to a new one) [46] of the selected model was conducted. For this purpose, we tested the model by using test sets obtained from six other patients (Nos. 3, 4, 11, 13, 19, and 21), randomly selected from the FSPEEG database. 

The raw data of the preictal, ictal, and interictal signals of these new patients were processed in the same way as those of patient No.17. For the sake of clarity, no specific feature selection was performed on these patients, but the calculated features were reduced using the selection rules obtained from patient No.17, obtaining the same 22 final features.

The model trained on patient No.17 failed to correctly detect seizures in patients 3, 11, 13, 19, and 21, although each patient responded differently to NN(20,7) (Table 6). Furthermore, the tests carried out showed a high number of false positives in the interictal periods of these five patients. On the other hand, one patient (No.4) seemed to respond quite well to NN(20,7). Therefore, the seizures of patient No.4 were successfully detected by the k-NN algorithm, using IR=20 and k=7, although, also in this case, the classifier showed a high FP number (12,484 elements wrongly classified as positive on 99,011 negative elements).

## 5. Discussion

In this paper, we propose an EEG-based automated seizure detection method, trying to investigate, qualitatively and quantitatively, its performance in terms of classification accuracy and, in particular, in terms of the number of false alarms, in order to understand how to reduce them. Our results show that, with our method, it is possible to recognize epileptic seizures while avoiding or dramatically minimizing the number of false alarms.

These results are at the same level as or at even higher level than those obtained with the application of more complex techniques, such as support vector machines or artificial neural networks [39], used for seizure detection.

In our study, the selection of two parameters assumed a key role:
IR relates to the class imbalance ratio. To overcome the problem of the unbalanced class, the elements of the majority class in the training set were undersampled by a procedure that we called OMU (Section 3.3). The more balanced the training set (corresponding to a decrease in IR values) was, the more positive elements were correctly classified. This also corresponded to an increase in the number of false positives; k, an internal parameter of the k-NN algorithm, represents the number of the nearest neighbors to be considered for the classification. A similar rule to that of the IR was also found for k, so there was a relationship between them; the more k and IR increased, the more the correct classifications of the negative elements increased, but the more correctly classified positive elements decreased.


One of the findings of k-NNs was that, once the value of k was fixed, the number of true positives increased as IR decreased. Therefore, the more balanced the training set was, the more k-NN correctly classified the positive examples of IKTAL, but also the greater the number of false positives was, and the lower the number of true negatives was. 

The feature-selection step also played an important role in our study for the following reasons:
It eliminated any redundancy of the variables. For example, the Joint Entropy (JE) and the Conditional Entropy (CE) calculated with respect to method B (in which the second signal was the null vector) were equal to the Shannon Entropy (SH). This equivalence is noticeable in their mathematical formulation and was discovered thanks to the PCC (for these features, the PCC is equal to 1). Moreover, some other extracted variables could be closely cross-related;It produced new knowledge or confirmed old knowledge. The three OutFokus electrodes (E4, E5, E6) were discarded by the feature selection filters, and only the InFokus electrodes computed in bands B40 (40–70 Hz) and B70 (70–120 Hz) survived the analysis. The selected features mainly came from the information theory (CE, JE, KC, and MI), while the others from the Seismic Evaluators category (SP) and the Distance-based category (LD);It improved, in some cases, the performances of the models compared with other methods where more features are analyzed by learning algorithms [39,40].


The number of false positives of the model was very low, but if we consider that it never classified more than two consecutive seconds as false positives, it is reasonable to assume that the number of false alarms was zero, since it is very rare to find a seizure which lasts less than two seconds [47]. Furthermore, since the minimum duration of the seizures of patient No.17 was 51 s, we can reasonably assume that no false seizures were detected by the selected k-NN. In this way, a threshold-based rule (in seconds) could be derived from the model in such a way that if the model detected a seizure lasting less than the threshold, then there was no real seizure in the EEG signal. In this study, we could set the threshold to 3 s, but a more conservative rule could also be considered to find a threshold that could be consistent across different patients.

Furthermore, the inspection of the original signal in order to understand how it was in the specific time points where we found the false positives revealed that the original signal in those time periods was most likely affected by artifacts, as shown in Figure 13. Thus, we think that adding an artifact-removal step in the preprocessing phase could reduce or even eliminate the number of false alarms.

We also investigated the possibility of applying the achieved detection model to other patients from the FSPEEG database. We found that, with the exception of a single patient (No.4), the model showed worse performance on other patients, demonstrating that our model is not exportable. Furthermore, even in patient No.4, despite the fact that the model showed a very high ability of correctly identifying the seizures (98.3% of correctly classified positive elements), an increase in the number of false positives was found. However, these two patients showed some common characteristics, and this result gives us reason to explore clusters of similar patients, for which it would be possible to train a unique model. This similarity shown by these two patients could be investigated more deeply from a clinical and / or neurological point of view. For example, an immediate consideration is that both patients had the same type of seizures (Temporal), and the average seizure duration was very similar. These findings push us to perform further research, by calculating, for example, the similarities among patients through an analysis of complexity metrics [48,49]. Moreover, the performance of the model on other patients was highly influenced by the feature-selection phase, which was conducted on patient No.17, so it was very patient specific. In addition, in order to perform a less restrictive feature-selection phase, it could certainly be very useful to conduct a preliminary study to discover groups of similar patients in order to be able to only export the detection model to patients belonging to the same cluster.

## 6. Conclusions, Study Limitation, and Future Works

The proposed model for seizure detection showed excellent results in detecting seizures, providing a very low false-positive rate during the interictal phase when tested on one patient. However, these results were not confirmed, except for one patient, when this model was tested on six other patients. Of course, this was mainly due to the fact that the model was trained on one specific patient; thus, the rules extracted for that patient were not exportable across all the patients. This limits the generalization ability of the model, making it difficult to be applied to different patients. This limitation could be addressed in the future by training the model using a group of patients who show some similar neurological characteristics (for example, the type of epilepsy or the average duration of seizures). As the performances of machine learning approaches increase as the available data increase, future tests could include a larger number of patients, thus a larger number of seizures. Furthermore, future works could include the application of clustering techniques in order to find similar patients, for whom it would be possible to train unique and general models. Another important contribution of this work is the analysis of the number of false alarms, which is a critical index for assessing an automated seizure detection method. Our qualitative analysis showed that the model never produced more than two seconds of “misdetected” seizures and that the false positives detected by the model were most likely due to the presence of artifacts in the signal; therefore, in the future, we aim to also include in the preprocessing step an artifact-removal technique, which could probably reduce or, ideally, zero the number of false alarms. In conclusion, although the proposed method could be considered patient specific, the availability of larger datasets, and the use of clustering techniques and of additional preprocessing steps could improve the generalization ability of the method, making it feasible to be applied to different kinds of patients and different types of epilepsy.

## Figures and Tables

**Figure 1 biomedicines-10-01491-f001:**
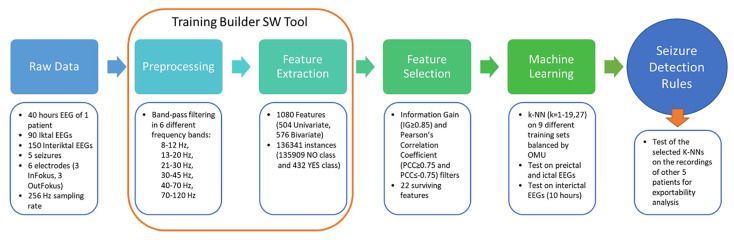
Pipeline of data analysis: from raw data to seizure detection rules.

**Figure 2 biomedicines-10-01491-f002:**
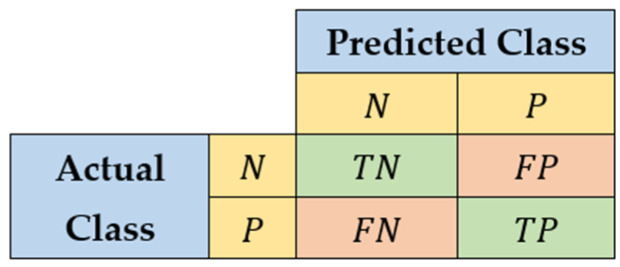
Binary confusion matrix.

**Figure 3 biomedicines-10-01491-f003:**
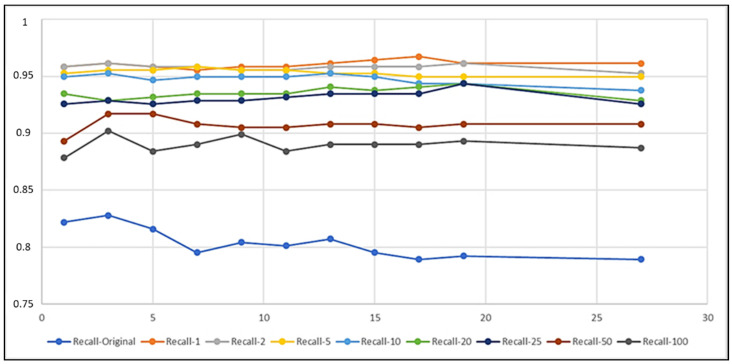
Recall–IR on Y (TPR) as k varied.

**Figure 4 biomedicines-10-01491-f004:**
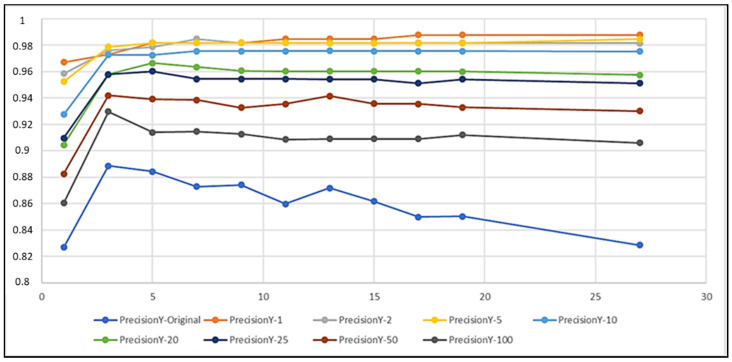
Precision–IR on Y as k varied.

**Figure 5 biomedicines-10-01491-f005:**
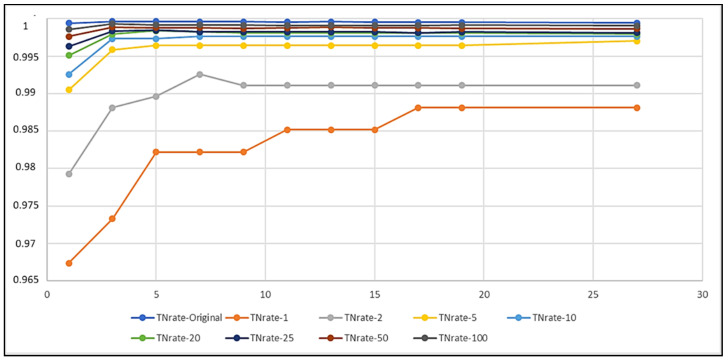
TNR–IR (Recall on N) as k varied.

**Figure 6 biomedicines-10-01491-f006:**
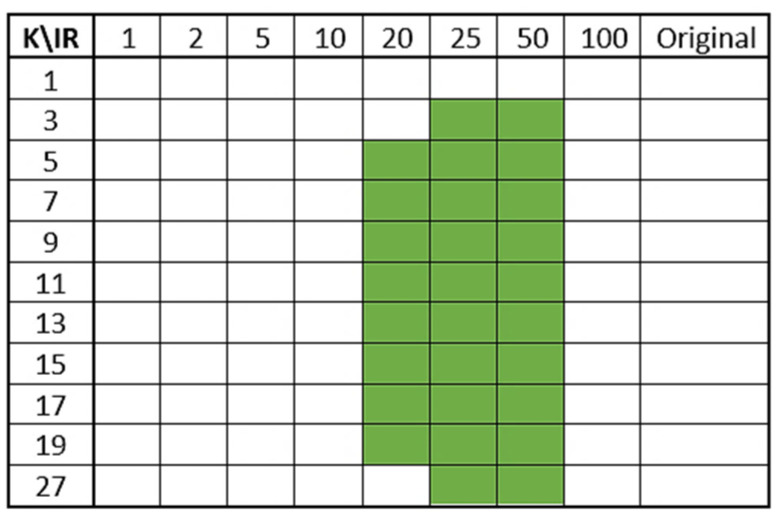
Best models selected by 10-fold cross-validation.

**Figure 7 biomedicines-10-01491-f007:**
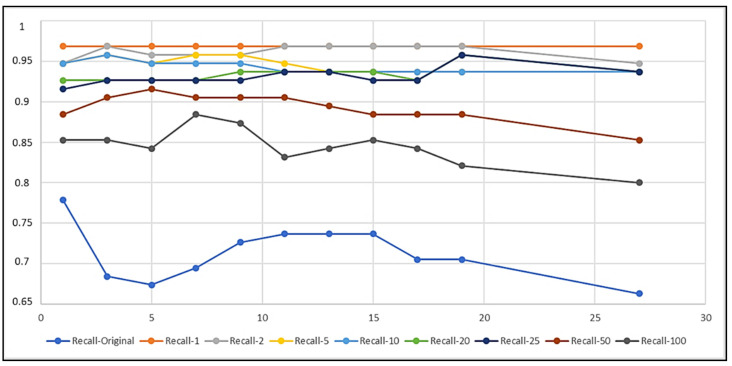
Curves to compare the results on the IKTAL set.

**Figure 8 biomedicines-10-01491-f008:**
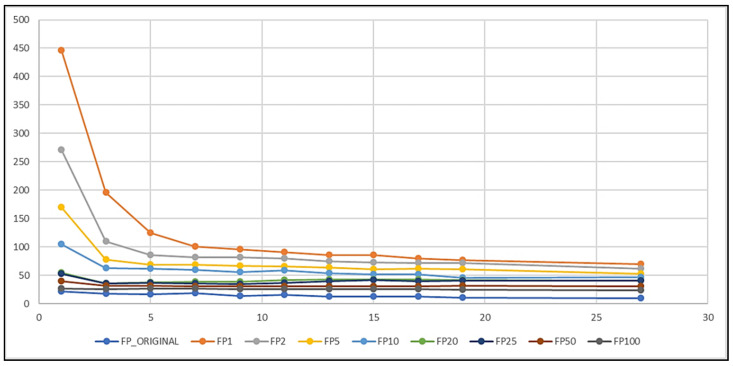
Curves to compare the results on the INTERIKTAL set.

**Figure 9 biomedicines-10-01491-f009:**
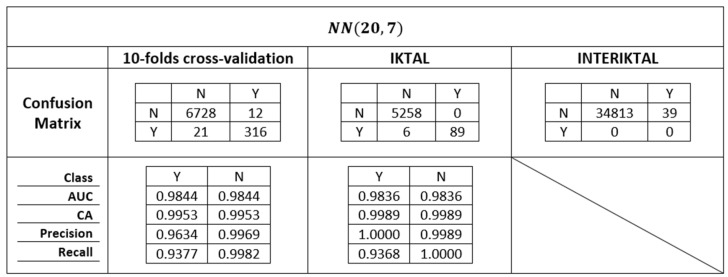
All performance metrics of final NN(20,7) model.

**Figure 10 biomedicines-10-01491-f010:**
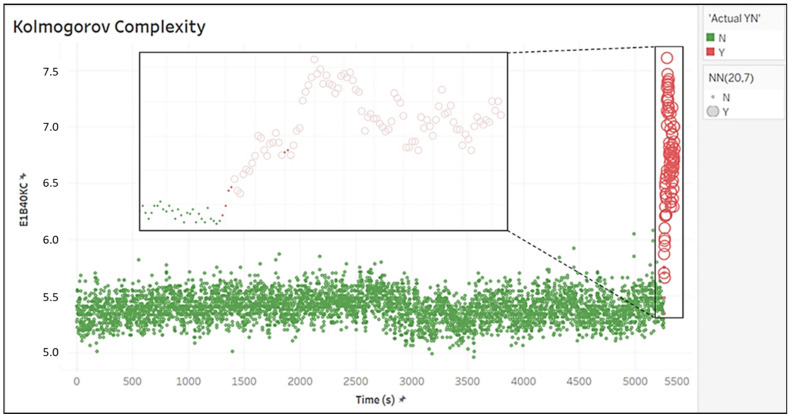
Time variation in the Kolmogorov Complexity in the IKTAL set and focus on errors in the classification by NN(20,7).

**Figure 11 biomedicines-10-01491-f011:**
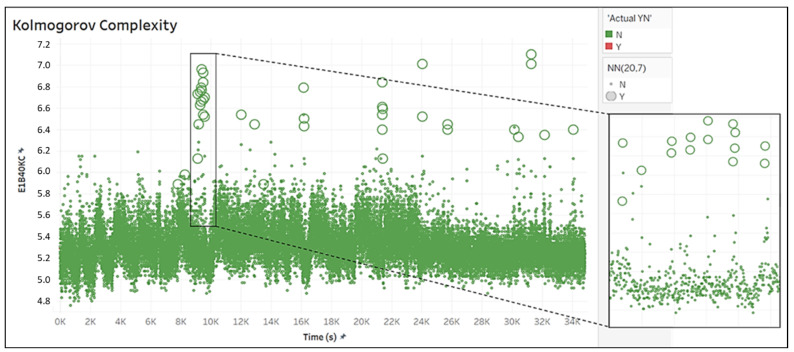
Time variation in the Kolmogorov Complexity in the INTERIKTAL set and focus on errors in the classification by NN(20,7).

**Figure 12 biomedicines-10-01491-f012:**
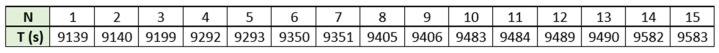
Non-consecutive seconds wrongly tagged with “Y” by NN(20,7).

**Figure 13 biomedicines-10-01491-f013:**
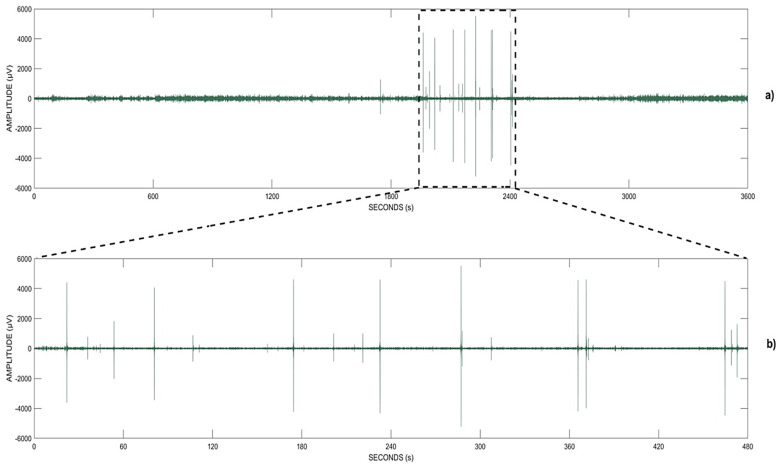
Qualitative analysis of false positives described in Figure 12: one-hour EEG signal from E1, band B40 (**a**), and a focused view on the time interval where the false positives were detected by the method (**b**).

**Table 1 biomedicines-10-01491-t001:** Features extracted by the Training Builder tool.

ID	Name of Feature	Code	U/B	Count
1	Kolmogorov Complexity	KC	U	36
2	Log Energy Entropy	LE	U	36
3	Lower-Limit Lempel-Ziv Complexity	LL	U	36
4	Upper-Limit Lempel-Ziv Complexity	LU	U	36
5	Shannon Entropy	SH	U	36
6	Averaged Period	AP	U	36
7	Inverted Time to Peak	IP	U	36
8	Peak Displacement	PD	U	36
9	Predominant Period	PP	U	36
10	Squared Grade	SG	U	36
11	Squared Time to Peak	SP	U	36
12	Hjorth Mobility	HM	U	36
13	Kurtosis	KU	U	36
14	Standard Deviation	SD	U	36
15	Cross Correlation Index	CC	B	36 MA + 36 MB
16	Conditional Entropy	CE	B	36 MA + 36 MB
17	Dynamic Time Warping	DT	B	36 MA + 36 MB
18	Euclidean Distance	ED	B	36 MA + 36 MB
19	Joint Entropy	JE	B	36 MA + 36 MB
20	Longest Common Sub-Sequence	LC	B	36 MA + 36 MB
21	Levenshtein Distance	LD	B	36 MA + 36 MB
22	Mutual Information	MI	B	36 MA + 36 MB

**Table 2 biomedicines-10-01491-t002:** Performance metrics.

#	Symbol	Performance Metric	Definition as	What Does It Measure?
			**Basic Measures**	
1	TPR	True-Positive Rate—Recall on P	$\frac{TP}{TP+FN}$	How good a model is at correctly predicting positive cases
2	TNR	True-Negative Rate—Recall on N	$\frac{TN}{TN+FP}$	How good a model is at correctly predicting negative cases
3	FPR	False-Positive Rate—Fall-out	$\frac{FP}{FP+TN}$	Proportion of incorrectly classified negative cases
4	PPV	Positive Predictive Value—Precision on P	$\frac{TP}{TP+FP}$	Proportion of correctly classified positive cases out of total positive predictions
5	NPV	Negative Predictive Value—Precision on N	$\frac{TN}{TN+FN}$	Proportion of correctly classified negative cases out of total negative predictions
			**Derived Measure**	
6	MCC	Matthews Correlation Coefficient	$\frac{TP\cdot TN-FP\cdot FN}{\sqrt{\left(TP+FP\right)\left(TP+FN\right)\left(TN+FP\right)\left(TN+FN\right)}}$	Correlation between observed and predicted classifications
			**Graphical Measure**	
7	AUC	ROC Area	Area Under the ROC Curve	Area under the plot of the TPR against the FPR

**Table 3 biomedicines-10-01491-t003:** Final features selected by filters.

**Electrode**	E1 (10)	E2 (8)	E3 (4)				
**Band**	B40 (6)	B70 (16)					
**Feature Name**	CE-MA (2)	CE-MB (2)	JE-MB (3)	KC (3)	LD-MA (3)	MI-MA (4)	SP (5)

**Table 4 biomedicines-10-01491-t004:** Description of the sets for data analysis.

*N*	Name	Description	Registration Numbers	Use
1	FULL (D)	*D* was achieved by preprocessing about 40 h of EEG signals of the interictal, preictal, and ictal phases of patient No.17 using TrB, which calculated 1080 features and the binary target class “Actual *YN*”. *D* had 136,341 instances, 135,909 tagged with “*N*” and 432 tagged with “*Y*”.	52–76 98–100 109–11 114–116 122–124 131–133	This dataset was used in the feature-selection phase by applying filters based on the Information Gain formula and the Pearson Correlation index.
2	FULL_SELECTED	The features were selected, achieving 22 final features + “Actual YN” target class.		Disjointed sets IKTAL, INTERIKTAL and T were obtained. FULL_SELECTED=T ∪ IKTAL ∪ INTERIKTAL.
3	ORIGINAL TRAINING (T)	Dataset with 96,135 instances, 95,798 tagged with “*N*” and 337 tagged with “*Y*”, coming from 4 preictal, 4 ictal, and 15 interictal phases.	52–60 71–76 98–100 114–116 124 131–133	This dataset was used for model training. The parameters of the *k*-*NN* algorithm were chosen by considering the 10-fold cross-validation.
4	IKTAL	About 1.5 h of preictal phase followed by a seizure of almost 95 s. This dataset had 5353 instances, 5258 tagged with “*N*” label and 95 tagged with “*Y*”.	122–123	This dataset was used for testing the selected models in order to maximize the number of correctly classified positive instances, thus to detect the seizure.
5	INTERIKTAL	About 10 h of records of interictal phase. All the 34,853 instances in this set were tagged with “*N*”.	61–70	This dataset was used for testing the models in order to reduce the number of false positives.

**Table 5 biomedicines-10-01491-t005:** MCC s of the 9 selected training sets. Each MCC was calculated as a maximum of 10 MCC values, and each of these was obtained by a 1 -NN trained on a different IR -undersampled set of T.

** *IR* **	1	2	5	10	20	25	50	100	284.3
** *MCC* **	0.949	0.951	0.955	0.954	0.955	0.954	0.936	0.925	0.865

**Table 6 biomedicines-10-01491-t006:** Results of NN(20,7) on other patients.

PAT	No. of Seizures	No. of Instances	Origin	Average Seizure Duration (s)	AUC	Recall on *P*	Recall on *N*
17	5	136,341	Temporal	86.16	-	-	-
3	5	108,836	Frontal	92.66	0.8411	0.5991	0.9826
4	4	99,363	Temporal	87.39	0.9835	0.9831	0.8739
11	3	103,520	Parietal	195.86	0.5992	0.0633	0.9992
13	2	95,127	Temporal/Occipital	158.28	0.5715	0.0221	0.9871
19	3	120,819	Frontal	12.54	0.5838	0.1672	1.0000
21	5	124,774	Temporal	89.09	0.5000	0.0000	0.9990

## Data Availability

The datasets generated during and/or analyzed during the current study are available from the corresponding author upon reasonable request.

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
