# Peer review of "Machine Learning Characterization of Ictal and Interictal States in EEG Aimed at Automated Seizure Detection"

_biomedicines, 2022, doi:10.3390/biomedicines10071491_

Round 1

Reviewer 1 Report

The study is interesting but the English needs professional editing.

Line 48: "often is" should be "is often"

Line 87: delete "by using"

Line 151: Rewrite the sentence from "which consider...". It is not correct grammatically.

Table 2: FPR should also be listed as a symbol.

Line 272: Rewrite the sentence. It does not make sense.

Line 491: Rewrite the sentence "we used as ..."

Line508: Delete "deeply"

Line 520: Make "but also false positives increase" as a independent sentence.

Line 562: Rewrite the sentence. 

Author Response

We thank the reviewer for the comments. Please find attached our point-by-point reply.

Reviewer 2 Report

The overall contents of this manuscript is not well organized to give a clear overview of this work. Author presented only one patient data. I have suggested some major comments and suggestions about this study are as the following:

Comments to the Authors:

  1. Authors should re-write the abstract of this study clearly with including objective, method, results, conclusion and significance.
  2. Authors should add more patients data at least 10 epilepsy patients to test this model.
  3. The introduction of this study is looking very weak. It should be divided into three paragraphs like background, gap between previous research to new develop method and hypothesis with specific aims.
  4. There is no conclusion section in this paper. Authors should write conclusion of this study clearly.
  1. My suggestion is that the authors should write some limitations of this model and future application.

Author Response

(The authors gave the same response as above.)

Round 2

Reviewer 2 Report

I have checked the revise version of this manuscript. I do not have any further comments. All my suggestions has been addressed by the authors.